# Ganoderic Acids Prevent Renal Ischemia Reperfusion Injury by Inhibiting Inflammation and Apoptosis

**DOI:** 10.3390/ijms221910229

**Published:** 2021-09-23

**Authors:** Guangying Shao, Jinzhao He, Jia Meng, Ang Ma, Xiaoqiang Geng, Shun Zhang, Zhiwei Qiu, Dongmei Lin, Min Li, Hong Zhou, Shuqian Lin, Baoxue Yang

**Affiliations:** 1State Key Laboratory of Natural and Biomimetic Drugs, Department of Pharmacology, School of Basic Medical Sciences, Peking University, Beijing 100191, China; 18210466583@163.com (G.S.); 18200288030@163.com (J.H.); mengjia_2011@pku.edu.cn (J.M.); maangxmu@outlook.com (A.M.); gengxq92@163.com (X.G.); tcsnjiu@163.com (S.Z.); 1410305218@pku.edu.cn (Z.Q.); leemin@bjmu.edu.cn (M.L.); rainbow_zhou@126.com (H.Z.); 2Fuzhou Institute of Green Valley Bio-Pharm Technology, Fuzhou 350002, China; lzxjuncao@163.com (D.L.); linshuqian2020@163.com (S.L.); 3JUNCAO Technology Research Institute, Fujian Agriculture and Forestry University, Fuzhou 350002, China; 4Key Laboratory of Molecular Cardiovascular Sciences, Ministry of Education, Beijing 100816, China

**Keywords:** ganoderic acids, renal ischemia reperfusion injury, acute kidney injury, inflammation, cell apoptosis, hypoxia reoxygenation

## Abstract

Renal ischemia reperfusion injury (RIRI) is one of the main causes of acute kidney injury (AKI), which can lead to acute renal failure. The development of RIRI is so complicated that it involves many factors such as inflammatory response, oxidative stress and cell apoptosis. Ganoderic acids (GAs), as one of the main pharmacological components of *Ganoderma lucidum*, have been reported to possess anti-inflammatory, antioxidant, and other pharmacological effects. The study is aimed to investigate the protective effect of GAs on RIRI and explore related underlying mechanisms. The mechanisms involved were assessed by a mouse RIRI model and a hypoxia/reoxygenation model. Compared with sham-operated group, renal dysfunction and morphological damages were relieved markedly in GAs-pretreatment group. GAs pretreatment could reduce the production of pro-inflammatory factors such as IL-6, COX-2 and iNOS induced by RIRI through inhibiting TLR4/MyD88/NF-kB signaling pathway. Furthermore, GAs reduced cell apoptosis via the decrease of the ratios of cleaved caspase-8 and cleaved caspase-3. The experimental results suggest that GAs prevent RIRI by alleviating tissue inflammation and apoptosis and might be developed as a candidate drug for preventing RIRI-induced AKI.

## 1. Introduction

Ischemia reperfusion injury (IRI) is characterized by the obstruction of the blood flowing to the particular organ, followed by the recovery of blood flow and reoxygenation, causing greater organ damage [1,2]. Kidney is one of the organs sensitive to IRI due to the particularity of tissue function and structure and its high demand for oxygen [3]. Renal ischemia reperfusion injury (RIRI) usually occurs after sepsis, renal transplantation and cardiac surgery. RIRI is one of the leading causes of acute kidney injury (AKI) [4,5], which often leads to high mortality in hospitalized patients [6]. However, there is still a lack of effective drugs to prevent or treat RIRI clinically [7].

According to existing studies, the pathogenic factors of RIRI are numerous and complex, and they are known to include but not be limited to calcium overload caused by imbalance of ion channels, excessive production of ROS, inflammatory response [8], protein kinase activation, endoplasmic reticulum and mitochondrial dysfunction, cell apoptosis and other interactions among different mechanisms [9,10]. Renal tubular cells are more susceptible to RIRI, and renal tubular epithelial cells can secrete cytokines and chemokines in response to pathogen stimulation [11]. At the late stage of reperfusion, inflammatory cytokines released from damaged cells bind to receptors on renal tubular cells. The ligand–receptor interaction forms a vicious circle of inflammatory cascades, leading to further damage to tubular cells [12]. Inflammatory response is triggered quickly after injury and then continues and even can be amplified throughout the whole course of the disease, followed by execution of tubular cells apoptosis. Previous studies have found that various anti-inflammatory drugs can alleviate RIRI to some extent, such as TLR4 blockers and complement inhibitors [13,14].

*Ganoderma lucidum* (*G. lucidum*), as the highly regarded traditional Chinese medicine, is widely used as medicine and functional food at present. Pharmacological experiments and clinical studies show that *G. lucidum* has a variety of pharmacological effects. Two major active components are responsible: *G. lucidum* polysaccharides and *G. lucidum* triterpenoids (GTs) [15]. As one of the most abundant GTs, ganoderic acids (GAs) have anti-inflammatory, anti-oxidative stress, anti-tumor and anti-apoptotic effects [16,17,18,19]. It has been reported that Ganoderic Acid F, one component of GAs, can suppress LPS-induced the activation of NF-kB pathway and inhibit LPS-triggered iNOS expression and the production and secretion of relative inflammatory cytokines in mice [16]. Therefore, we hypothesize that GAs can alleviate RIRI via inhibiting inflammatory response and cell apoptosis.

The study aimed to investigate the preventive effect of GAs on RIRI and its underlying mechanisms. The experiments involved a mouse model of RIRI and a cell model of hypoxia/reoxygenation. The experimental results demonstrated that GAs pretreatment could reduce the inflammatory response through inhibiting TLR4/MyD88/NF-kB signaling pathway and alleviate cell apoptosis, which suggests that GAs have the potential to be developed as a preventive agent for preventing RIRI-induced AKI.

## 2. Results

### 2.1. GAs Protected the Kidney against RIRI

Firstly, the levels of blood urea nitrogen and blood creatinine were measured to explore the effects of GAs on renal function. Both parameters of mice undergoing renal ischemia reperfusion operation were higher than those of sham mice. Administration of GAs at the dose of 1.7, 5, 15 mg/kg before ischemia and reperfusion operation significantly prevented from increased blood urea nitrogen and creatinine induced by RIRI (Figure 1A,B). The morphological changes of renal tissues were analyzed by paraffin section and hematoxylin and eosin (H&E) staining. Compared with sham mice, we found proximal tubular damages including tubular brush border loss, renal tubular luminal dilatation and intracellular fragments in the ischemia reperfusion group, which were attenuated by GAs pretreatment (Figure 1C,D). These experimental results suggest that GAs could prevent from RIRI-induced renal injury to a significant degree.

### 2.2. GAs Had No Significant Effect on Oxidative Stress Induced by RIRI

Considering the improvement effect of GAs on renal structure and function, the mechanisms involved were worth further investigation. Oxidative stress plays a key role in the pathogenesis of RIRI. Free radical accumulation acts as an endogenous danger signal, and then can induce renal tubular epithelial cells to secrete cytokines and chemokines subsequently in response. Based on the results of the preliminary experiment, reperfusion for 24 h was chosen as the early time point of injury to determine the level of oxidative stress of RIRI and anti-oxidative stress effect of the GAs on RIRI, while reperfusion for 48 h was considered as the time point when the inflammatory response was more pronounced. Compared with sham mice, superoxide dismutase (SOD) and glutathione (GSH), which have antioxidant activity, were decreased in model groups. Administration of GAs had no significant effect on the levels of SOD and GSH, as shown in Figure 2A,B. In addition, as probably the most commonly used DNA damage marker for oxidative stress, 8-hydroxydeoxyguanosine (8-OHdG) was measured. Compared with sham mice, proximal tubular damages and strong 8-OHdG immunoreactivity were observed in RIRI-induced renal tissues. The damages of renal structure were attenuated by GAs pretreatment but no visible reduction in the immunostaining of 8-OHdG was observed in RIRI + GAs group (Figure 2C). The result of immunohistochemical staining for 8-OHdG was consistent with SOD and GSH changes. These experimental results suggest that the inhibitory role of GAs on RIRI was not primarily through suppressing oxidative stress.

### 2.3. GAs Inhibited RIRI-Induced Inflammatory Response

Apart from oxidative stress, several pro-inflammatory cytokines and mediators have been reported to play important roles in RIRI-induced AKI [20]. IL-6, COX-2 and iNOS are classical inflammatory mediators and can be activated in RIRI-induced inflammatory response. Therefore, the protein levels of IL-6, COX-2 and iNOS were measured to investigate the effect of GAs on inflammatory response. As shown in Figure 3A, the levels of three inflammatory mediators were all upregulated in RIRI-challenged mice and could be reversed by GAs pretreatment. Western blot results indicate that GAs could alleviate the inflammatory response activated by RIRI.

To explore the underlying mechanism of GAs in inhibiting RIRI-induced inflammation, we studied the effect of GAs on the activation of the NF-kB and its upstream signaling pathways in renal tissues, which are closely related to the abnormal release of pro-inflammatory factors in inflammatory response [21]. After recognizing endogenous danger signals such as oxygen free radicals and dead cell fragments, TLR4 then can trigger the intracellular association with its adapter molecules (MyD88) and activate a series of transcription factors (such as NF-kB) to promote the innate immune response [22,23,24]. Western blot (Figure 3B) analysis revealed that GAs suppressed the activity of TLR4 and MyD88 induced by RIRI. The experimental results suggest the involvement of the TLR4-dependent MyD88-NF-kB signaling pathway during GAs-mediated anti-inflammatory effects on RIRI.

### 2.4. GAs Ameliorated Apoptosis in the RIRI Model

Cell apoptosis plays a crucial role in RIRI due to the fact that apoptotic cell death occurs during RIRI and the degree of apoptosis can directly reflect the severity of the injury. The level of cell apoptosis induced by RIRI was evaluated visually by a TUNEL assay. We found more TUNEL-positive cells appeared in model kidneys than in sham kidneys. GAs pretreatment also reduced RIRI-induced apoptotic cells (Figure 4A,B), which suggests that GAs protect kidneys from renal tubular cell apoptosis. In addition, Western blot analysis showed that ratios of cleaved caspase-3/caspase-3 and cleaved caspase-8/caspase-8 were decreased significantly in GAs pretreated-model renal tissues, compared with RIRI-induced renal tissues (Figure 4C). These data indicate that GAs have an anti-apoptotic effect in the RIRI model.

### 2.5. GAs Suppressed H/R-Induced Inflammatory Response in NRK-52E Cell

H/R model was performed on NRK-52E cell to mimic RIRI in vitro. A CCK-8 assay was applied to measure the cytotoxicity of GAs at concentrations from 0 g/mL to 200 g/mL. Even at the maximum concentration, GAs showed no obvious cytotoxicity to NRK-52E cells. To verify the anti-inflammatory effect of GAs in vitro and the relationship between dose and efficacy, we designed three doses of GAs (3.125, 12.5, 50 g/mL) for the cell model. H/R model augmented inflammatory response in NRK-52E cell as evidenced by increased protein levels of IL-6, COX-2 and iNOS. GAs co-incubation administration significantly ameliorated inflammatory response in H/R-challenged groups by restoring the factors mentioned above (Figure 5A).

We further tested the changes of TLR4-MyD88-NF-kB signaling pathway. Similar to in vivo data, the protein expression of TLR4, MyD88 and phosphorylated NF-kB were higher in the H/R group than that in the control group, which was markedly suppressed by GAs (Figure 5B). These data indicate that GAs can ameliorate H/R-induced inflammatory response in a dose-dependent manner.

### 2.6. GAs Reduced Cell Apoptosis Induced by H/R

To examine the anti-apoptotic effect of GAs on H/R model, we performed a TUNEL assay. H/R injury markedly increased the ratio of apoptotic cells in comparison with the control group. The apoptosis was alleviated by GAs, which was dependent on the concentration of GAs (Figure 6A).

Previous studies have shown that active caspase-8 can induce the final enzymatic cascades of apoptosis by caspase-3 as the initiator of exogenous apoptosis [25]. The decrease of activity of caspase-8 and caspase-3 in GAs treatment group was proven by the reduction in the intensity of the band (Figure 6B). These results suggest that GAs depress cell apoptosis induced by H/R by restraining the activation of caspases.

## 3. Discussion

Recently it was reported that GAs had anti-inflammatory [18] and anti-apoptotic effects. However, the protective effect of GAs on RIRI-induced AKI and the underlying pathogenetic mechanisms have not been further investigated. Here, an in vivo model and an in vitro model were performed to simulate RIRI clinically and showed that GAs could relieve renal injury via down regulating TLR4/MyD88/NF-kB signaling pathway and reducing cell apoptosis.

A RIRI model was designed to follow our previous studies [7,26]. Transcriptome analysis of RIRI illustrated that the transcription reprogramming related genes gradually increase from the early stage of the injury to the later stage [27,28]. Previous studies suggest that RIRI is mainly divided into two stages: ischemia and reperfusion. At the initial stage of RIRI, the excessive enrichment of free radicals leads to vascular endothelial injury and the enrichment of free radicals further causes the release of inflammatory response factors and apoptosis [20,29]. The pathological data of renal ischemia reperfusion injury suggest that the inflammatory cascade amplifies in the middle and late stages of injury, so we conducted a preliminary experiment to explore the experimental conditions for confirming the peak time of inflammatory response and chose 24 h as the appropriate duration of reperfusion for the assessment of oxidative stress and 48 h for inflammatory response. Subsequent results showed the increased levels of oxidative stress after reperfusion for 24 h and more secretion of pro-inflammatory factors after reperfusion for 48 h. Correspondingly, we measured the levels of inflammatory cytokines at different time points after reoxygenation in vitro, and confirmed 12 h as the experimental condition according to the time with the strongest inflammatory activation in the H/R model in vitro. The selection of different reperfusion time points to study the pathogenesis and pharmacological effects benefits the clinical surgical strategy of RIRI and the future application of GAs.

During the process of RIRI, total damage is attributable to both ischemia phase and reperfusion phase, which means the injury of ischemia with reperfusion is more severe than ischemia-related AKI alone. When ischemia occurs, tissue is starved of oxygen and cells switch to anaerobic metabolism to supply ATP, which leads to acidosis and inadequate ATP production. Meanwhile, active Ca^2+^ efflux and Ca^2+^ reuptake by the endoplasmic reticulum reuptake are significantly reduced, leading to intracellular Ca^2+^ overload. During reperfusion, oxygen is supplied again and ATP is produced, which promotes additional Ca^2+^ influx and ROS production, and activates a cascade of inflammation and thrombosis, eventually exacerbating cellular damage [20]. Oxidative stress, inflammatory response, cell apoptosis and other mechanisms are all related. Especially the inflammatory response is triggered after injury and then throughout the whole course of the disease [30]. Not only damaged renal tubular epithelial cells secrete cytokines and chemokines, but also interaction of other signaling pathways lead to the release of pro-inflammatory factors [31]. For example, migrating mast cells and macrophages secrete inflammatory factors (TNF-a, IL-2, IL-6, IL-8), monocyte chemotactic factor (MCP-1) and other chemical chemokines to participate in inflammation, causing further aggravating cell damage [32,33].

Stimulating signals, such as cells undergoing ischemic necrosis and cell membrane rupture, known as the damage-associated molecular patterns (DAMPs), can be recognized by pattern recognition receptors (PRRs) [34]. TLR is an important type of PRP, which locates on the surface of cell membrane. When binding to DAMP, TLR can activate a series of transcription factors (such as NF-kB) to promote the innate immune response, which often participates in the inflammatory response [35,36]. TLR4 is even involved in regulating the processes of the infiltration of leukocytes in the kidney during ischemic acute kidney injury and the release of IL-6 [37]. We found that the TLR4/MyD88/NF-kB pathway forms a signal axis and participates in inflammatory response. GAs pretreatment contributed to decrease expression of TLR4 and its central adaptor protein, MyD88, to downregulate the downstream signaling pathways.

IL-6 is one of important inflammatory cytokines in the pathogenesis of AKI, which can be synthesized by endothelial cells under the stimulation of hypoxia [38]. Clinical studies have shown that the clearance of blood IL-6 and TNF-a by continuous renal replacement therapy can improve the prognosis of patients with AKI [39], suggesting that inflammation plays an important role in the development of AKI. Further studies showed that treatment with IL-6 antagonists reduced the apoptosis of renal tubular epithelial cells induced by RIRI to improve renal function and renal pathology [40,41]. These studies confirm the important role of inflammatory factors such as IL-6 in the development and treatment of RIRI. Both in vivo and in vitro experiments demonstrated that GAs pretreatment depressed the level of IL-6 and COX-2, suggesting the potential effect of inhibiting inflammatory response and restoring renal function in response to RIRI.

Cell apoptosis occurs via several pathways during RIRI-induced AKI, including the intrinsic pathway, extrinsic pathway and crosstalk between the above two pathways [42]. In the extrinsic apoptotic pathway, caspase-8 activation can be mediated by death receptors. Furthermore, activated caspase-8 also induces the intrinsic pathway by cleaving Bid to truncated Bid. The two pathways initiate the final enzymatic cascades of apoptosis by caspase-3 [43]. Western blot analysis exhibits decreased ratios of cleaved caspase-3/caspase-3 and cleaved caspase-8/caspase-8 in pretreatment group, suggesting the anti-apoptotic effect of GAs on RIRI. Thus, we put forward a possible mechanism hypothesis in which GAs prevent the kidneys from RIRI by reducing inflammatory response and cell apoptosis (Figure 7). The diagram illustrates that RIRI simulation leads to the activation of TLR4 and then TLR4 binds to the adaptor molecule MyD88 and activates IKK-IkB-NF-kB by a series of cascade reactions, subsequently synthesizing and releasing IL-6, COX-2 and iNOS. Moreover, TLR4 can also propagate the initial injury to transmit the apoptotic signals. Activation of the inflammatory response and apoptotic signaling pathway can be suppressed by GAs.

Moreover, GAs include three main monomers, GA-A, GA-B, GA-C2. Wan, B. et al. found that GA-A decreased proinflammatory cytokines in the bronchoalveolar lavage fluid in LPS-induced lung injury [44]. For cyclophosphamide-induced hepatotoxicity, GA-A could alleviate liver injury and hepatocyte apoptosis via inhibiting the activation of Txnip/Trx/NF-κB pathway [45]. GA-B also exerted anti-inflammatory effect in LPS-induced pneumonia, which is related to the regulation of Rho/NF-κB signaling pathway [46]. In addition, GA-C2 was found to significantly inhibit DNA fragmentation and decrease caspase activity to inhibit cell apoptosis in a-AMA-induced liver injury. Referring to the current literature available, none of the monomers above have shown significant cytotoxicity. However, compared to other monomers, GA-A showed the strongest inhibitory activity in renal cyst growth [47] and renal fibrosis [48] in vivo. According to the content of GA-A in GAs and its pharmacological activity in other experimental models, all of these monomers may play renoprotective roles in RIRI, but GA-A has the strongest pharmacological activity and is the easiest to obtain, which will be focused on in following studies.

## 4. Materials and Methods

### 4.1. Ganoderic Acids Preparation

GAs are hazel-colored powders extracted and isolated from Ganoderma lucidum, kindly provided by Fuzhou Institute of Green Valley Bio-Pharm Technology. In the in vivo experiment, GAs were dissolved in saline with 5% tween 80 and in the in vitro experiment, GAs were dissolved in DMSO for cell incubation. GAs were separated and prepared by water extraction and alcohol precipitation method from the extract-like product precipitated at the bottom of concentration tank during water extraction of Ganoderma lucidum. Then the crude samples were obtained by using the preparation of Flash medium pressure. According to the results of HPLC, three main monomers, GA-A, GA-B, GA-C2, account for 16.1%, 10.6% and 5.4% of crude GAs, respectively. The purity of these three monomers was >98%, as determined by HPLC [49,50].

### 4.2. Ethics Statement

This study was carried out in strict accordance with the recommendations of the Guide for the Care and Use of Laboratory Animals of China Association for Laboratory Animal Science. All animal care protocols were approved by the Institutional Animal Care and Use Committee at the Peking University Health Science Center (Beijing, China, LA220354, 19 May 2020). All sacrifice was performed under pentobarbitone anesthesia, and every effort was made to minimize animal suffering.

### 4.3. RIRI Mouse Model

Male C57BL/6J mice (8–10 weeks old) weighing 20–23 g were purchased from the Animal Center of Peking University Health Science Center. The male mice were fed in a 12/12 h light/dark cycle and free access to food and water. The mice were assigned to six groups of 6–8 animals for each condition randomly: vehicle control (5% tween 80 dissolved in saline)-sham group; GAs-sham group; vehicle control-RIRI group; GAs-IR group with three doses (1.7, 5, or 15 mg· kg^−1^ ·day^−1^). To make the warm RIRI model, the mice were anesthetized by intraperitoneal injections of sodium pentobarbital (80 mg/kg). The right renal artery of the mouse was clamped for 35 min with a small vascular clamp and the left kidney was removed. In the process of renal ischemia, the clamped kidney gradually darkened in color. Additionally, reperfusion stage was determined by observing the color changes of the kidney. The same surgical procedure carried out in the vehicle control-sham and GAs-sham group animals, except for the occlusion of the renal arteries.

For evaluating the prevention efficacy of GAs against RIRI, GAs were given by daily intraperitoneal injection at a concentration of 1.7, 5, or 15 mg· kg^−1^ ·day^−1^ for 3 days before the procedure until sacrifice. Blood and kidney samples after reperfusion for 1 or 2 days were collected for the assessment of oxidative stress and inflammatory response, respectively.

### 4.4. Blood BUN and Creatinine Measurement

Whole blood was centrifuged at 3000 rpm for 15 min at 4 °C to obtain serum to determine the levels of blood BUN by a QuantiChrom Urea Assay kit (BioAssay Systems, Hayward, CA, USA) and creatinine by with commercial kits (NJJC Bio, Nanjing, China) under the guidance of the manufacturer’s instructions.

### 4.5. Measurement of GSH and SOD

The kidney samples were homogenized with 0.9% (*w*/*v*) ice-cold physiological saline for 10 s and the supernatant was obtained after centrifuging at 12,000 rpm for 15 min at 4 °C. The activities of GSH and SOD in homogenized kidney samples were tested using specific assay kits (NJJC Bio, Nanjing, China) according to the manufacturer’s instructions.

### 4.6. Hematoxylin-Eosin Staining and Terminal Deoxynucleotidyl Transferase-Mediated 2′-Deoxyuridine 5′-Triphosphate Nick-End Labeling (TUNEL) Assay

After sacrifice, the renal tissues were fixed in 4% paraformaldehyde in 4 °C for paraffin embedding and sectioning. Sections were cut 3 mm thick and stained with hematoxylin and eosin for light microscope examination. Paraffin sections of kidney tissue or 6-well plates were prepared for TUNEL analysis using the One Step TUNEL Apoptosis Assay Kit (FITC) (Meilunbio, Dalian, China), following the manufacturer’s protocol.

The pictures were captured using a fluorescence microscope and the number of TUNEL-positive cells was counted under ×200 microscopic fields. The apoptotic index was calculated as (number of apoptotic cells/total number of nucleated cells) × 100. At least 3 random fields were counted for each group.

### 4.7. Cell Culture

The NRK-52E cells (rat renal proximal tubule epithelial cells) were purchased from the Cell Resource Center of Shanghai Institutes for Biological Sciences, Chinese Academy of Sciences (Shanghai, China). NRK-52E cells were cultured in DMEM containing 10% fetal bovine serum (FBS; Gibco, Melbourne, Australia), 2 mM glutamine, 100 U/mL penicillin and 100 μg/mL streptomycin, in a humidified atmosphere with 5% CO_2_ at 37 °C. Cells were seeded into 100 mm diameter culture dishes at a concentration of 6 × 10^4^ cells/mL.

### 4.8. Cell Viability Assay

The CCK-8 assay kit (Dojindo, Rockville, MD, USA) was employed for testing cytotoxicity of GAs in vitro. NRK-52E cells were planted in 96-well plates at a density of 5000 cells/well. GAs was co-cultured with NRK-52E cells for 12 h before measurement. Then, the CCK-8 solution, at a 1/10 dilution with DMEM medium containing 10% FBS, was added to each well, and the cells were incubated for 2 h at 37 °C. Absorbance at 450 nm was measured with a microplate reader (Biotek, MQX200, Winooski, VT, USA). Cell viability was calculated as follows:Cell viability (%) = (OD_treatment_ − OD_blank_)/(OD _control_ − OD _blank_) × 100

### 4.9. Cell Hypoxia/Reoxygenation (H/R)

In general, 60~70% confluent NRK-52E cells were selected for the study and then they were serum-deprived for 24 h. GAs were added to the cell cultures for 12 h before the procedure at a concentration of 3.125 mg/mL, 12.5 mg/mL, 50 mg/mL. During the hypoxia process, DMEM media was replaced by starving low-glucose DMEM under low-oxygen (1% O_2_) for 12 h in a humidified hypoxia incubator (Thermo Fisher Scientific, Waltham, MA, USA). Then the cells were incubated in fresh DMEM and exposed to normal oxygen (95% air + 5% CO_2_) for 12 h. Control groups were incubated in normal conditions (fresh DMEM containing 10% FBS and normal oxygen) and changed the DMEM at the same time as the model groups during the H/R process.

### 4.10. Western Blot Analysis

Renal tissues or cells were homogenized in RIPA lysis buffer (Mei5bio) containing 4% protease inhibitor cocktail (Roche, Basel, Switzerland) and 1% phosphatase inhibitor (Applygen, Beijing, China). Total protein concentration was detected by BCA kit (Pierce, Rockford, IL, USA) after ultrasonic cracking. Identical amounts of protein samples were electrophoresed on polyacrylamide gels and electrotransferred to polyvinylidene difluoride membranes, and then the membranes were incubated with antibodies against b-actin (Santa Cruz), COX-2, IL-6, TLR4 (ABclonal, Wuhan, China), iNOS, MyD88 (Abcam, Cambridge, UK), caspase-3, cleaved caspase-3, caspase-8, cleaved caspase-8, NF-kB, p-NF-kB (Immunoway, Plano, TX, USA), at 4 °C overnight. Then, goat anti-rabbit IgG or goat anti-mouse IgG (EASYBIO, Beijing, China) were added and the blots were developed with ECL plus kit (Biodragon, Beijing, China) and were visualized with a chemiluminescence detection system (Syngene, GeneGnome XRQ, Cambridge, UK). Quantitation was performed by scanning and analyzing the intensity of the hybridization bands and the data were analyzed with Image J software.

### 4.11. Statistical Analyses

All results were expressed as means ± SEM by GraphPad Prism 7.0 software. A two-tailed Student’s *t*-test was used to analyze two sets of data and one-way ANOVA was used for multiple sets of data. A *p*-value of <0.05 was considered to be statistically significant.

## 5. Conclusions

In summary, our experimental results showed that GAs could prevent RIRI-induced AKI by inhibiting the activation of TLR4 and MyD88, which further prevent the initiation of downstream inflammatory response and cell apoptosis pathway, eventually improving renal function for the first time. This study will be a clue to understanding the pathogenesis of the acute kidney injury, especially in relation to RIRI, and GAs could have the potential to be developed as a candidate drug for RIRI-induced AKI clinically. However, our work still has some limitations and remains to be elaborated in the future research. Furthermore, the comprehensive and specific mechanisms involved in RIRI are still unclear at present, which needs to be updated in the near future.

## Figures and Tables

**Figure 1 ijms-22-10229-f001:**
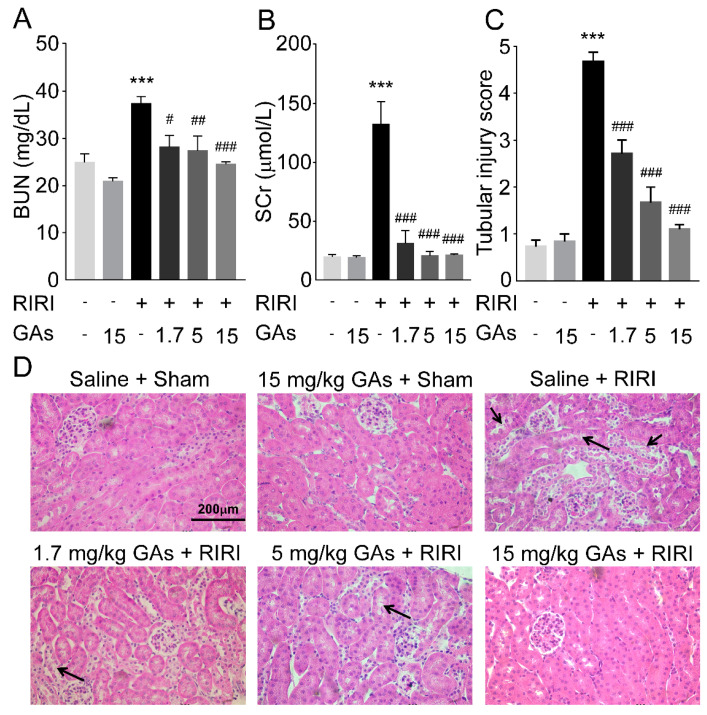
GAs protected the kidney against RIRI in mice. (**A**) Blood urea nitrogen. (**B**) Serum creatinine. Values are expressed as the mean ± SEM (*n* = 6~8). (**C**) Quantification of tubular injury scores (*n* = 8). (**D**) Representative H&E staining of renal tissue. (400× original magnification, typical proximal tubular damages are indicated by black arrows). *** *p* < 0.001 compared with Saline + Sham group; ^#^
*p* < 0.05, ^##^
*p* < 0.01 and ^###^
*p* < 0.001 compared with Saline + RIRI group (two-tailed Student’s *t*-test).

**Figure 2 ijms-22-10229-f002:**
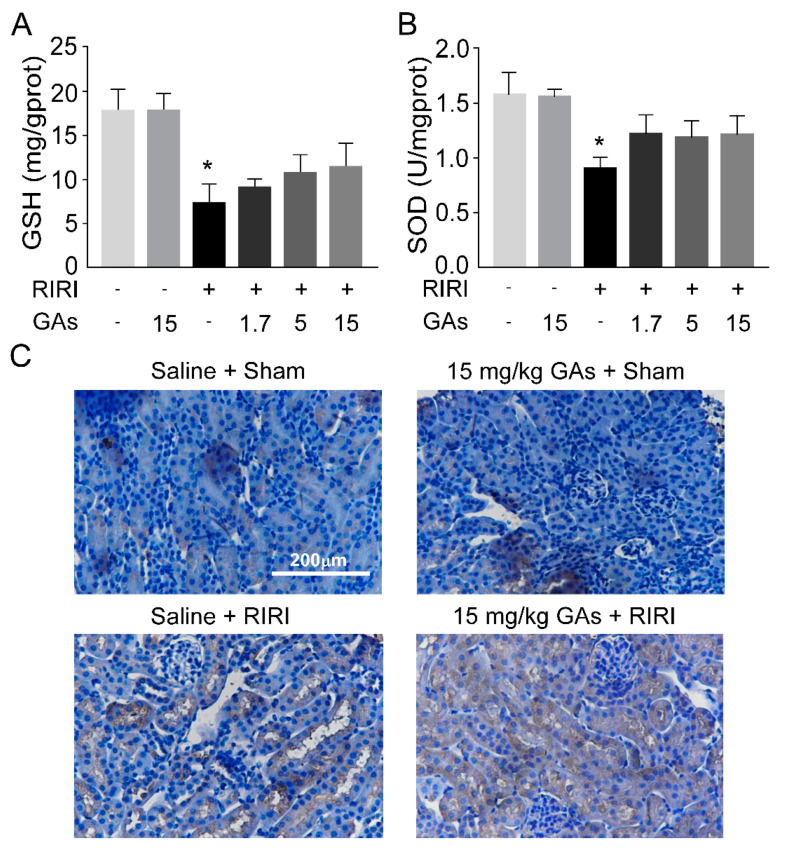
GAs attenuated RIRI-induced renal injury not primarily through suppressing oxidative stress. (**A**) GSH. (**B**) SOD. (**C**) Immunohistochemical localization of 8-OHdG. Values are expressed as the mean ± SEM (*n* = 6). * *p* < 0.05 compared with Saline + Sham group (two-tailed Student’s *t*-test or one-way ANOVA).

**Figure 3 ijms-22-10229-f003:**
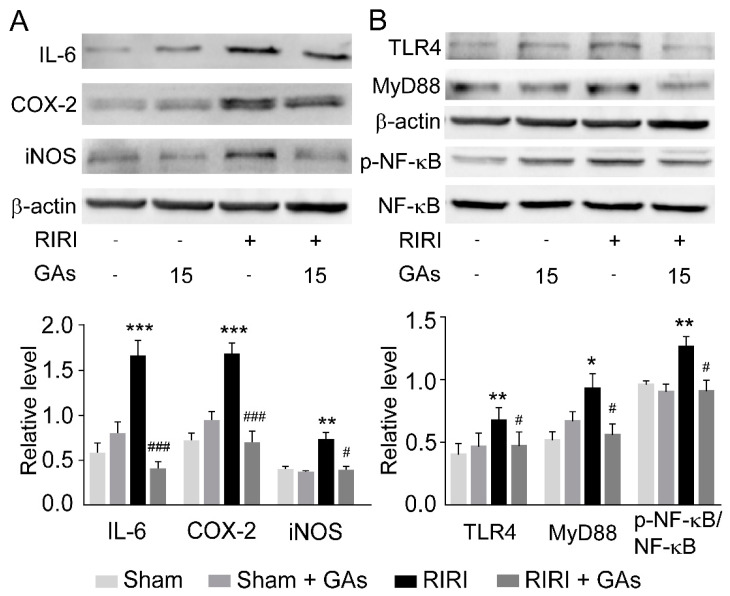
GAs attenuated RIRI-induced renal injury by suppressing inflammatory response. (**A**) Representative Western blots and the relative protein levels of IL-6, COX-2, and iNOS in the kidney. (**B**) Representative Western blots and the relative protein levels of TLR4, MyD88, and p-NF-kB in the kidney. The data of target proteins were normalized to b-actin as internal reference and are expressed relative to the intensity of the Saline + Sham group. Values are expressed as the mean ± SEM (*n* = 4–6). * *p* < 0.05, ** *p* < 0.01 and *** *p* < 0.001 compared with Saline + Sham group; ^#^
*p* < 0.05 and ^###^
*p* < 0.001 compared with Saline + RIRI group (two-tailed Student’s *t*-test or one-way ANOVA).

**Figure 4 ijms-22-10229-f004:**
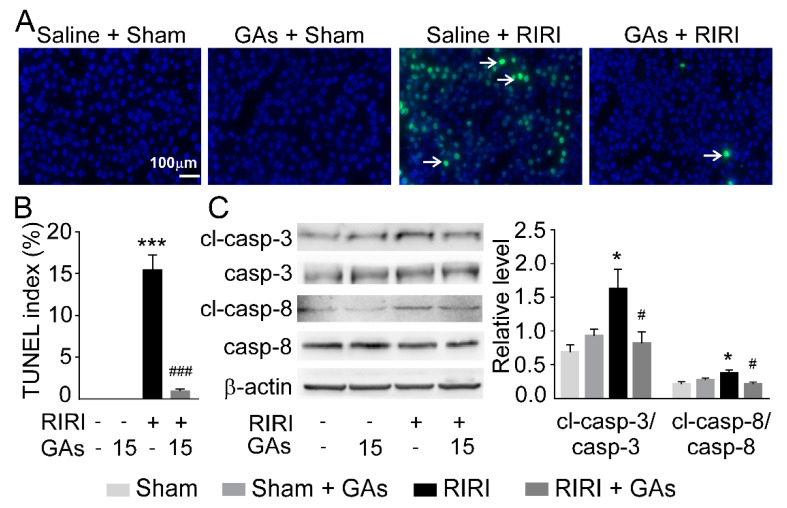
GAs inhibited RIRI-mediated apoptosis in mice. (**A**) Representative TUNEL staining (green fluorescence, magnification 200×) and (**B**) TUNEL-positive index (*n* = 8). (**C**) Representative Western blots and the relative protein levels of cleaved caspase-3, caspase-3, cleaved caspase-8 and caspase-8 in the kidney. The data of target proteins are expressed relative to the intensity of the Saline + Sham group. Values are expressed as the mean ± SEM (*n* = 6). * *p* < 0.05 and *** *p* < 0.001 compared with Saline + Sham group; ^#^ *p* < 0.05 and ^###^ *p* < 0.001 compared with Saline + RIRI group (two-tailed Student’s *t*-test or one-way ANOVA).

**Figure 5 ijms-22-10229-f005:**
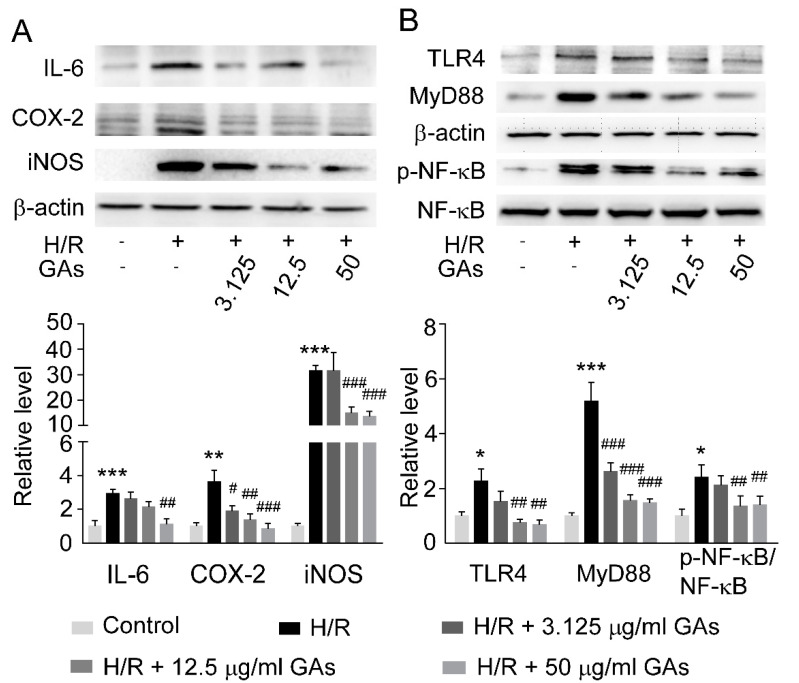
GAs inhibited H/R-mediated inflammation by adjusting TLR4/MyD88/NF-kB signaling in NRK-52E cell. (**A**) Representative Western blots and quantitative densitometric analysis of inflammation-related proteins in NRK-52E cell. (**B**) Representative Western blots and quantitative densitometric analysis of TLR4, MyD88, NF-kB and p-NF-kB in NRK-52E cell. The data of target proteins were normalized to b-actin as internal reference and are expressed relative to the intensity of the Control group. Values are expressed as the mean ± SEM (*n* = 5–6). * *p* < 0.05, ** *p* < 0.01 and *** *p* < 0.001 compared with Control group; ^#^
*p* < 0.05, ^##^
*p* < 0.01 and ^###^
*p* < 0.001 compared with H/R group (two-tailed Student’s *t*-test or one-way ANOVA).

**Figure 6 ijms-22-10229-f006:**
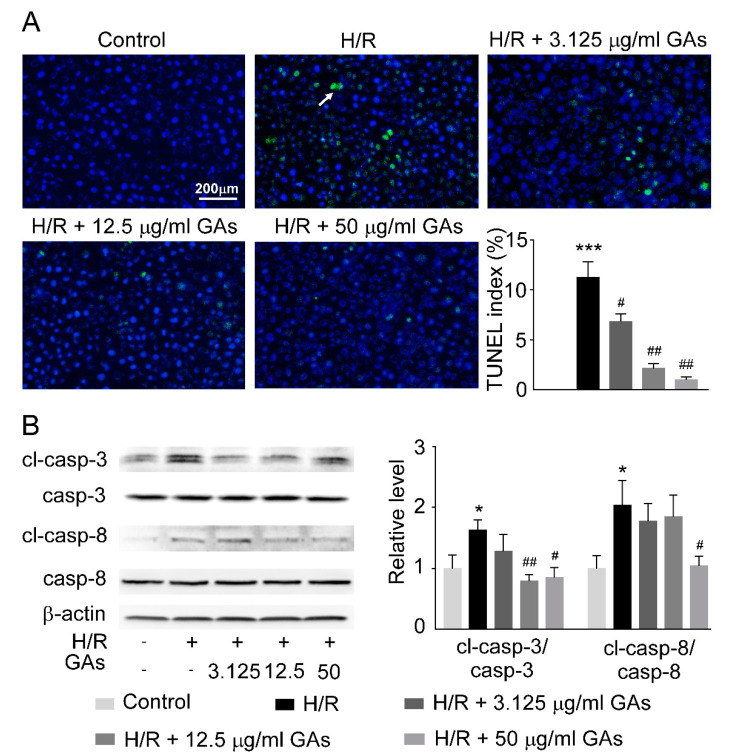
GAs suppressed the activation of apoptosis in response to H/R in NRK-52E cell. (**A**) Representative TUNEL staining (magnification 200×) and the quantification of TUNEL-positive index (*n* = 8). (**B**) Representative Western blots and the relative protein levels of key enzymes involved in apoptosis in NRK-52E cell. The data of target proteins are expressed relative to the intensity of the Control group. Values are expressed as the mean ± SEM (*n* = 5~6). * *p* < 0.05 and *** *p* < 0.001 compared with Control group; ^#^
*p* < 0.05 and ^##^
*p* < 0.01 compared with H/R group (two-tailed Student’s *t*-test or one-way ANOVA).

**Figure 7 ijms-22-10229-f007:**
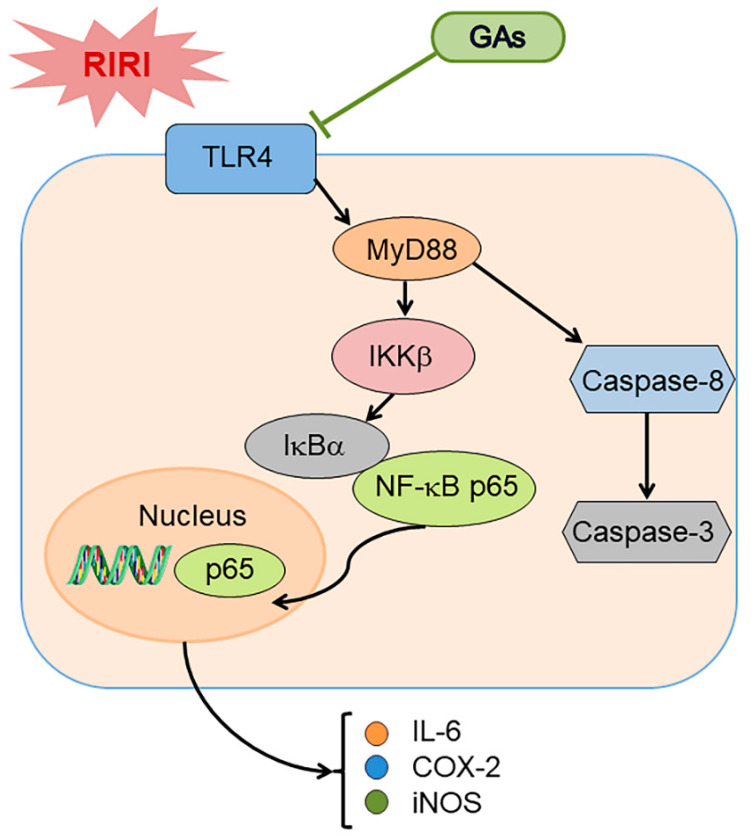
Schematic diagram of the signal pathways involved in RIRI-induced renal injury and proposed effects of GAs on RIRI. Please see the text for explanations.

## Data Availability

All data generated or analyzed during this study are included in this article and are available from the corresponding author on reasonable request.

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
