# Peer review of "Ganoderic Acids Prevent Renal Ischemia Reperfusion Injury by Inhibiting Inflammation and Apoptosis"

_ijms, 2021, doi:10.3390/ijms221910229_

Round 1
Reviewer 1 Report
The authors investigated via experimental work the role Ganoderic acids play in the prevention of renal ischemia reperfusion injury by the inhibition of inflammation and apoptosis.
The manuscript is well written and of scientific value. I would suggest some minor recommendations:
1)to focus more on the apoptotic mechanism that is triggered by ischemia. This could be for example reporting on the consequences in the renal tubules (after transplantation)
2)considering the impact of GA on short ischemic times versus prolonged
3)the role played by GA in the necrosis
Author Response
Response to Reviewer 1 Comments
The authors investigated via experimental work the role Ganoderic acids play in the prevention of renal ischemia reperfusion injury by the inhibition of inflammation and apoptosis.
The manuscript is well written and of scientific value. I would suggest some minor recommendations:
Point 1:to focus more on the apoptotic mechanism that is triggered by ischemia. This could be for example reporting on the consequences in the renal tubules (after transplantation)
Response 1: Clinically, renal ischemia reperfusion injury (RIRI) occurs in kidney transplantation and renal trauma, which includes both ischemia and reperfusion processes. Cell apoptosis during RIRI is caused by the combined action of ischemia and reperfusion. For example, during ischemia, tissue oxygen content decreases, while during reperfusion, oxygen free radicals explode, thus inducing cell apoptosis. We chose RIRI over just ischemia as the model for pharmacological study of GAs. We will further study the pathogenesis of ischemia and reperfusion respectively, which will be helpful to further understand the pathogenesis of RIRI.
Point 2: considering the impact of GA on short ischemic times versus prolonged
Response 2:As you suggested, the injury degree of RIRI in mice increases with the prolongation of ischemia time. Minor renal injury caused by IRI is reversible, and renal function can gradually return to normal after acute stage. Severe renal injury can significantly increase the mortality of mice and affect renal function for a long time. To better assess the effect of GAs on RIRI, we selected 35 minutes as the ischemia duration to detect mouse kidney injury, which simulates clinical renal ischemia reperfusion injury and the model has been previously validated in our laboratory.
Point 3: the role played by GA in the necrosis
Response 3: Several different cell death programs are induced in response to RIRI stimuli together, such as apoptosis, necrosis and so on. According to “The term apoptosis was first referred to in relation to renal I/R, it has been implicated as an eminent symptom of I/R-induced organ damage”, we mainly focused on the apoptotic pathway.
At present, various pharmacological inhibitors like necrostatins (RIPK1 inhibitors, necroptosis), ferrostatins (ferroptosis), sanglifehrin A (MPT-associated death) and olaparib (parthanatos) have been used to target cell death and have been investigated in animal models. The safety of strategies for inhibiting non-apoptotic cell death pathways still need to be confirmed. We will pay attention to the pharmacological effect of GAs against necrosis later. Thank you for your suggestions, which are very helpful.
Reviewer 2 Report
This is well presented paper and the results are significant within the field. The introductory story goes very well and it points well to hypothesis and aims. Results are well presented and well discussed. All the necessary figures and tables are presented in the manuscript and are supporting evidence for the findings described in this paper. Material and methods were sufficiently explained.
Line 17, Ganoderma lucidum – G should be capital letter
Line 54-55, Chinese medicine instead of Chinese medicines
Line 294, in vitro should be italicized
Can you discuss a bit more about monomers of GAs and their cytotoxic, apoptotic and anti-inflammatory activities from previous literature available? What do you think are these monomers acting synergistically or the activity is related to one monomer? Please refer to current literature available.
Please add more points to the conclusion section. Why this study is so important, what is its significance? What are further steps that need to be undertaken? – You can transfer the last paragraph from discussion section here. What have you shown for the first time?
Author Response
Response to Reviewer 2 Comments
This is well presented paper and the results are significant within the field. The introductory story goes very well and it points well to hypothesis and aims. Results are well presented and well discussed. All the necessary figures and tables are presented in the manuscript and are supporting evidence for the findings described in this paper. Material and methods were sufficiently explained.
Point 1:Line 17, Ganoderma lucidum – G should be capital letter
Response 1: It has been corrected.
Point 2:Line 54-55, Chinese medicine instead of Chinese medicines
Response 2: It has been revised.
Point 3:Line 294, in vitro should be italicized
Response 3: It has been revised.
Point 4:Can you discuss a bit more about monomers of GAs and their cytotoxic, apoptotic and anti-inflammatory activities from previous literature available? What do you think are these monomers acting synergistically or the activity is related to one monomer? Please refer to current literature available.
Response 4: These are good suggestions. We have added the discussion in the manuscript.
Point 5:Please add more points to the conclusion section. Why this study is so important, what is its significance? What are further steps that need to be undertaken? – You can transfer the last paragraph from discussion section here. What have you shown for the first time?
Response 5: We have added these in the manuscript.